# In Silico Development of a Chimeric Multi-Epitope Vaccine Targeting *Helcococcus kunzii*: Coupling Subtractive Proteomics and Reverse Vaccinology for Vaccine Target Discovery

**DOI:** 10.3390/ph18091258

**Published:** 2025-08-25

**Authors:** Khaled S. Allemailem

**Affiliations:** Department of Medical Laboratories, College of Applied Medical Sciences, Qassim University, Buraydah 51452, Saudi Arabia; k.allemailem@qu.edu.sa; Tel.: +966-16-301-0555

**Keywords:** reverse vaccinology, subtractive proteomics, multi-epitope vaccine (MEV), *Helcococcus kunzii*, immunoinformatics, in silico vaccine design

## Abstract

**Background**: *Helcococcus kunzii*, a facultative anaerobe and Gram-positive coccus, has been documented as a cunning pathogen, mainly in immunocompromised individuals, as evidenced by recent clinical and microbiological reports. It has been associated with a variety of polymicrobial infections, comprising diabetic foot ulcers, prosthetic joint infections, osteomyelitis, endocarditis, and bloodstream infections. Despite its emerging clinical relevance, no licensed vaccine or targeted immunotherapy currently exists for *H. kunzii*, and its rising resistance to conventional antibiotics presents a growing public health concern. **Objectives**: In this study, we employed an integrated subtractive proteomics and immunoinformatics pipeline to design a multi-epitope subunit vaccine (MEV) candidate against *H. kunzii*. Initially, pan-proteome analysis identified non-redundant, essential, non-homologous, and virulent proteins suitable for therapeutic targeting. **Methods/Results**: From these, two highly conserved and surface-accessible proteins, cell division protein FtsZ and peptidoglycan glycosyltransferase FtsW, were selected as promising vaccine targets. Comprehensive epitope prediction identified nine cytotoxic T-lymphocyte (CTL), five helper T-lymphocyte (HTL), and two linear B-cell (LBL) epitopes, which were rationally assembled into a 397-amino-acid-long chimeric construct. The construct was designed using appropriate linkers and adjuvanted with the cholera toxin B (CTB) subunit (NCBI accession: AND74811.1) to enhance immunogenicity. Molecular docking and dynamics simulations revealed persistent and high-affinity ties amongst the MEV and essential immune receptors, indicating a durable ability to elicit an immune reaction. In silico immune dynamic simulations predicted vigorous B- and T-cell-mediated immune responses. Codon optimization and computer-aided cloning into the *E. coli* K12 host employing the pET-28a(+) vector suggested high translational efficiency and suitability for bacterial expression. **Conclusions**: Overall, this computationally designed MEV demonstrates favorable immunological and physicochemical properties, and presents a durable candidate for subsequent in vitro and in vivo validation against *H. kunzii*-associated infections.

## 1. Introduction

*Helcococcus kunzii*, an opportunistic, Gram-positive bacterium that exhibits facultative anaerobic characteristics, is attracting growing interest because of its part in clinical microbiology, especially with regard to immunocompromised patients [1]. It was considered in the past to be a harmless commensal germ that lives in the skin and mucous membrane, but this has changed since its discovery in countless medical disorders [2]. These health disorders include endocarditis, diabetic foot ulcers, bacteremia, necrotic fasciitis, and prosthetic joint infections [3]. The virulent *H. kunzii* is gaining more and more recognition, particularly in cases of polymicrobial infection, since a lot of case studies and clinical reports have been presented as evidence of its role in severe systemic infections [4]. Given the high level of its attestation in immunocompromised individuals, especially in people with diabetes mellitus, cancer, or implanted metallic medical devices, its reference in the novel pathogen category is justified, and its clinical attacks are perhaps now underestimated [2]. The prevalence of the increased awareness regarding the medical relevance of *H. kunzii* as an organism contrasts with the lack of vaccines or specifically immunotherapeutic tools despite the exigent need to address innovative and effective preventive methodologies [1]. It is found that the scarcity of vaccine candidates following conventional or computationally based predesigns against *H. kunzii* indicates a severe research gap in spite of increasing documentation of drug resistance and clinical applicability.

Synergistically with its increasing clinical relevance, *H. kunzii* has demonstrated resistance to commonly used antibiotics, thus affecting the outcome of patients and the effectiveness of medications [5]. Multiple research studies and case reports have proven variations in susceptibility to macrolides and clindamycin, as well as resistance to 2-lactam antibiotics, including ampicillin and penicillin, all of which are relatively uncommon and underreported [6]. This resistance is likely to be caused by the development of beta-lactamase enzymes or modifications of penicillin-binding proteins in *H. kunzii*, and this resistance compromises the effectiveness of antibiotics that halt cell wall synthesis [7].

Additionally, *H. kunzii* can form biofilms, especially in the polymicrobial setting, which will help it to persist in the tissues of the host and reduce its sensitivity to antimicrobial drugs, a critical challenge to the current treatment regimens [8]. Vaccinations are an imperative approach to reducing the morbidity burden of *H. kunzii* infection, since first-line antibiotics are becoming resistant and few options for treatment remain. These findings demonstrate the dire urgency of developing alternative therapeutic methods, such as vaccines, to responsively handle infections caused by this increasingly drug-resistant pathogen [9].

There are considerable constraints of the current methods of traditional vaccination, which include live-attenuated types of vaccination and vaccination with inactivated whole cells being inefficient in terms of immunogenicity, a possibility of reversion and pathogenicity, and adverse effects following administration of vaccines that are toxic or hypersensitive to non-specific antigenic components [10]. Moreover, single-protein subunit vaccines are prone to producing off-target side effects and provide only a limited level of immune protection, particularly against genetically heterologous pathogens [11]. In contrast, epitope-based vaccines, particularly multi-epitope vaccines (MEVs), possess multiple advantages by combining the most immunodominant and conserved B-cell and T-cell epitopes to produce highly targeted and long-lasting immune responses [12]. These MEVs are designed to minimize undesired side effects and provide high efficacy through targeting the most protective regions of pathogen antigens which are devoid of sequences and are allergenic or homologous to human proteins [13].

Identification of effective vaccine candidates has been significantly simplified by advances in proteomics and reverse vaccinology, particularly in the case of recently identified or little-understood diseases such as those related to *H. kunzii* [14]. To abbreviate the proteome to the most viable candidates for vaccine development, subtractive proteomics can be used to methodically eliminate non-surface-exposed proteins that are homologous to humans, or non-essential proteins [15]. This enhances specificity and decreases the likelihood of autoimmune response or cross-reactivity [14]. Conversely, reverse vaccinology is not in need of traditional in vitro culture since it employs the pathogen’s whole genome or proteome to computationally identify putative antigenic proteins [16]. When coupled with immunoinformatics tools, screening of candidate proteins and their corresponding epitopes can be systematically assessed for antigenicity, immunogenicity, and population coverage, thus greatly speeding up the vaccine development pipeline [17].

Immunoinformatics-guided strategies have already been effectively applied in MEV design against a widespread collection of pathogens, for instance viruses SARS-CoV-2, Dengue, and Zika, bacteria *Mycobacterium tuberculosis*, *Helicobacter pylori*, and *Klebsiella pneumoniae*, and protozoan parasites Leishmania and Plasmodium species [18,19,20,21,22,23,24]. Notably, a number of studies have extended beyond in silico testing to experimental testing, strengthening the translational potential of bioinformatics-based vaccine design. As an example, by employing a computational approach, Peele et al. [25] developed a SARS-CoV-2 MEV and validated its expression and immunogenicity with molecular and structural biology techniques. Liao et al. [26] recently reported the development and successful in vivo validation of an MEV against *Klebsiella pneumonia* using a reverse vaccinology pipeline. On the other hand, another research group previously applied reverse vaccinology to identify promising immunogenic and drug targets against antibiotic-resistant *Neisseria gonorrhoeae* [27]. These examples indicate the viability and practicability of the immunoinformatics approach in practice when developing vaccines. These approaches integrate epitope prediction, structural modeling, molecular docking, and immune simulation to develop vaccine candidates that are capable of encouraging strong cellular and humoral immune responses [24]. Taking these advancements into account, the present study develops a novel multi-epitope vaccine candidate for use against *H. kunzii* by combining subtractive proteomics, reverse vaccinology, and immunoinformatics. To the best of our knowledge, this study represents the first in silico design of a chimeric multi-epitope subunit vaccine specifically targeting *Helcococcus kunzii*, an emerging Gram-positive pathogen with increasing clinical significance and reported antibiotic resistance. This study addresses critical challenges such as antibiotic resistance and the lack of effective vaccines for this emerging pathogen. Conserved and essential proteins of *H. kunzii* were chosen in this study, applying a core proteome approach. Then, in silico prediction and assessment of high-antigenic-potential epitopes was performed. Following this, the epitopes were synthesized into a comprehensive multi-epitope construct, and applying molecular docking simulation and dynamic simulation, powerful computer systems were used to assess its ability to interact with a key human immune receptor (TLR4) as well as its structural stability and immunogenicity.

## 2. Results

### 2.1. Proteome Subtraction and Candidate Protein Selection

The full proteome of *H. kunzii* ATCC 51366 consists of 1876 proteins. Subtractive proteomics analysis on the Geptop 2.0 server identified 282 proteins as a requirement for the survival of the bacteria. BLASTp was conducted against the human proteome to decrease the chance of such autoimmune reactions at a minimum of 91 non-homologous essential proteins. Subcellular localization with the PSORTb server split these proteins, with 71 being cytoplasmic and 16 being associated with the membranes. Based on subcellular localization, antigenicity, and non-allergenicity (Table 1), two proteins, a cytoplasmic protein and a cytoplasmic membrane-associated protein, were selected as lead vaccine candidates. They were peptidoglycan glycosyltransferase FtsW, a required enzyme for peptidoglycan biosynthesis, important in the maintenance of bacterial cell wall integrity, and cell division protein FtsZ, a tubulin homolog that contributes to the initiation of septum formation in bacterial cytokinesis. For further evaluation of their vaccine potential, transmembrane helix prediction by TMHMM 2.0 indicated that both proteins lacked transmembrane helices, representing a favorable topology for vaccine design and the accessibility of the epitope.

### 2.2. Antigenicity Prediction

The antigenic capacity of the chosen proteins was assessed with the VaxiJen v2.0 server, which is an ACC transformation-based protective antigen prediction tool. For a threshold cutoff value of 0.5 for bacterial models, cell division protein FtsZ was predicted to be antigenic with a score of 0.5439, reflecting its capacity to induce an immune response. Equally, peptidoglycan glycosyl transferase FtsW also had antigenic behavior with a score of 0.5169. These findings corroborate the listing of both proteins as ideal candidates for subsequent vaccine design.

### 2.3. Epitope Selection Phase

To select antigenic proteins (FtsZ and FtsW) to forecast cytotoxic T-lymphocyte (CTL), helper T-lymphocyte (HTL), and linear B-cell (LBL) epitopes to engineer a robust and broad-spectrum multi-epitope subunit vaccine, the IEDB MHC Class I binding prediction tool was used to forecast 9-mer peptides with high binding capacity to a broad spectrum of HLA Class I alleles. The top CTL epitopes were chosen based on their high antigenicity, as predicted by VaxiJen v2.0, positive IEDB Class I immunogenicity tool immunogenicity scores, confirmation of non-allergenicity using AllerTOP v2.0, and non-toxicity employing ToxinPred2 tool. Peptides with tight binding affinities (i.e., peptides with scores less than 2.0) were chosen in a bid to identify strong lymphocyte (CTL) epitopes, since this threshold indicates a likelihood of having high chances to bind with HLA Class I alleles (Table 2). The IEDB MHC Class II binding prediction tool was applied to generate 15-mer peptide candidates that may bind to HLA Class II molecules in order to predict helper T-lymphocyte (HTL) epitopes. According to the IFNepitope server’s prediction of these HTLs’ ability to induce interferon-gamma (IFN-γ) responses, the top five were selected. Toxicity assessments, immunogenic potential, allergenicity, and antigenicity were other screening criteria. To ensure inclusion of highly promising candidates for CD4+ T-cell-mediated immunological activation, a percentile rank criterion was applied, and only epitopes with ranks higher than 2.0 were retained (Table 3). For linear B-cell epitopes, prediction was carried out on the ABCpred server with an epitope length of 16 mer and a threshold score value of 0.51 chosen to improve prediction accuracy. The epitopes were subsequently tested for allergenicity, antigenicity, and toxicity to confirm their suitability for inclusion in the final vaccine design (Table 4). This stringent selection process ensures that the final selected epitopes have high effectiveness in inducing a robust overall immune response.

### 2.4. Multi-Epitope Vaccine Construction: Integration of Adjuvants and Linkers for Synergistic Immune Activation

Immune activation can enhance immunogenicity and induce a robust immunological response; as such, the multi-epitope vaccine (MEV) in this study was carefully built by covalent coupling of T-cell and B-cell epitopes onto a suitable adjuvant. Through enhancement of overall immunological activity and activation of the innate immune system, adjuvants play a dynamic part in enhancing vaccine ability to boost the immunogenic activity of vaccine formulations. The cholera toxin subunit B (accession > AND74811.1) of Vibrio cholerae O1 biovar El Tor was selected as the adjuvant in the course of this study. Through the addition of this adjuvant, the immunostimulatory capacity of the vaccine is highly amplified, which is an important part of its overall effectiveness. A few were employed to secure the clean assembly of the epitopes with the adjuvant and retain their biological activity. The cytotoxic T-lymphocyte (CTL) epitopes were coupled to the adjuvant utilizing the EAAAK linker, which secured the spatial positioning essential for optimal immunological presentation and encouraged structural rigidity. In addition, CTL and helper T-lymphocyte (HTL) epitopes were intentionally joined by means of the AAY and GPGPG linkers, respectively. These linkers enable precise epitope presentation and help to firmly activate both arms of cellular immunity. Bi-lysine (KK) linkers were used to connect linear B-cell (LBL) epitopes. The KK linkers maintain the immunoreactive characteristics of B-cell epitopes so that they can bind with B-cell receptors and elicit a humoral antibody response. Linkers play a crucial role in maintaining the functional properties of B-cell epitopes, which are required to elicit a strong humoral immune response. The final multi-epitope vaccine (MEV) design consisted of 347 amino acids. Its components, which include epitopes, linkers, and the adjuvant, were carefully arranged structurally to ensure both conformational stability and enhanced immunogenic potential. Steric clashes were purposefully avoided in this design to facilitate optimal spatial presentation with the host immune system. Each epitope’s careful placement and covalent attachment are meant to improve cooperative immune pathway activation without sacrificing the epitopes’ individual effectiveness. As a result, the designed MEV is a promising vaccine candidate that can induce humoral and cellular immunity in a multi-epitope fashion, offering broad-spectrum protection against pathogenic threats (Figure 1).

### 2.5. Population Coverage Analysis of Selected CTL and HTL Epitopes

To assess the global significance of the chosen helper T-lymphocyte (HTL) and cytotoxic T-lymphocyte (CTL) epitopes included in the multi-epitope vaccine (MEV), a thorough population coverage analysis was conducted. According to the findings, the combined collection of epitopes would put over 74% of the global population at risk of developing immunological reactions. The highest expected coverage was surprisingly found in European populations, with France leading at 88%, closely followed by Sweden (87%), England (87%), and Germany (86%). This broad population reach illustrates how individuals with diverse genetic backgrounds may be protected by the vaccine design. The estimated 82% population coverage across Europe showed strong regional applicability. Outside of Europe, there was also notable coverage in several other nations, such as China (72%), Pakistan (69%), Japan (67%), and Russia (74%). These results highlight the broad-spectrum nature of the chosen epitopes and suggest that they are appropriate for developing a multi-epitope vaccination (MEV) that can protect genetically diverse people worldwide (Figure 2).

### 2.6. Post-Translational and Physicochemical Analysis of the Vaccine Construct

The physicochemical and structural characteristics of the final multi-epitope vaccine (MEV) construct were thoroughly examined using the ProtParam program in order to determine its expression viability, stability, and immunogenic efficacy. It is well suited for recombinant production and purification procedures, as evidenced by its predicted molecular weight of approximately 43.05 kDa. The construct may have mildly acidic properties that impact its solubility and behavior at physiological pH levels, as indicated by its calculated theoretical isoelectric point (pI) of 5.56. The balanced charge of the MEV construct distribution, which supports its structural integrity and water solubility, was demonstrated by the presence of 25 positively charged (lysine and arginine) and 31 negatively charged (glutamic acid and aspartic acid) residues. With an instability index of 30.11, the protein was classified as stable and appropriate for downstream production and long-term storage. Better thermostability and the capacity to maintain structural conformation under a range of temperature conditions are also suggested by the computed aliphatic index of 70.40. A key characteristic for successful immunological interaction and vaccine development is the protein’s hydrophilic tendency, which is indicated by its GRAVY score of −0.068. This makes the protein suitable for solubility in bodily fluids. The MEV construct is highly stably expressed across a variety of expression platforms, as evidenced by its expected half-life of about 30 h in mammalian reticulocytes, greater than 20 h in yeast, and exceeding 10 h in *E. coli* when tested for in vivo stability. Extensive assessments of allergenicity, toxicity, and antigenicity were conducted using AllerTOP v2.0, ToxinPred, and VaxiJen v2.0, correspondingly. The analysis verified the construct’s high antigenic qualities and lack of toxicity and allergy. These characteristics show the MEV’s safety profile and suggest that it might be a good candidate for additional preclinical research and experimental validation as a subunit vaccine.

### 2.7. Structural Analysis and Validation of the Multi-Epitope Vaccine Construct

In order to gain a full understanding of the structural organization of the proposed multi-epitope vaccine (MEV), secondary and tertiary structure studies were conducted. The MEV’s 397-residue sequence has 158 random coils (39.80%), 94 extended strands (23.68%), and 145 α-helices (36.52%), as predicted by the PSIPRED and SOPMA tools for secondary structure analysis. A well-balanced and ordered protein structure is reflected by the arrangement of secondary structural units, which is required for maintaining functional stability and for enhancing optimal immune recognition. To predict the tertiary structure, the 3D model of the vaccine construct was foreseen with the AlphaFold server tool, which employed cutting-edge deep learning methodologies for precise structural modeling. For improvement in the structural quality, the modeled structure was developed with the GalaxyRefine server, which enhances global and local accuracy by side-chain repacking and whole energy minimization. The structure obtained was then validated through Ramachandran plot analysis, indicating that 94.0% of residues occupied the most favorable regions, with 4.5% in the additionally allowed regions and only 0.6% in disallowed regions. This large percentage of residues within favorable conformational regions firmly endorses the accuracy of the predicted structure (Figure 3). Additional structural verification was executed utilizing ProSA-web (https://prosa.services.came.sbg.ac.at/prosa.php accessed on 11 April 2025) and ERRAT (https://saves.mbi.ucla.edu/ accessed on 12 April 2025) tools. The ERRAT examination provided a quality factor value of 86.550, signifying a high degree of structural correctness. The ProSA Z-score of −3.39 was comfortably within the limits of native proteins of similar size, verifying the lack of structural defects or misfolded segments (Figure 4). These findings collectively ensure that the optimized MEV construct has a stable, well-folded, and functionally pertinent tertiary structure and is a valid candidate for downstream experimental analysis.

### 2.8. Prediction and Selection of B-Cell Epitopes

B lymphocytes (B cells) are the backbone of the humoral constituent of the adaptive immune system, liable for neutralizing pathogens by producing antibodies. Hence, the inclusion of highly operative B-cell epitopes is important in inducing strong and specific antibody-mediated immunity in subunit vaccine development. In the current research, both conformational (discontinuous) and linear (continuous) B-cell epitopes were foreseen employing the ElliPro server, which uses protein 3D structural data to predict antibody-accessible areas. The three-dimensional architecture of the final vaccine construct was submitted to the ElliPro server. Epitope prediction was performed using default threshold settings—a minimum binding score of 0.5 and a distance cutoff of 6 Ångströms—to ensure the detection of epitopes with high accessibility and structural flexibility. With these parameters, 18 conformational B-cell epitopes were detected, varying in length from 3 to 30 amino acid residues, with prediction scores reaching from 0.57 to 0.967, indicating their possible immunogenic significance. In addition, 10 linear B-cell epitopes were also foreseen as enriching the epitope map towards maximal humoral activation. These conformational epitopes were then visualized through PyMOL v1.3 for spatial validation of their surface exposure as well as spatial distribution in the 3D vaccine construct (Figure 5). The inclusion of conformational as well as linear B-cell epitopes plays a vital role in the width and strength of the predicted immune response, maximizing the overall effectiveness and immunogenicity of the multi-epitope vaccine candidate.

### 2.9. Molecular Docking Analysis with Host Immune Receptor

Molecular docking is a core computational method for the prediction and assessment of binding attraction and interface kinetics between an immunizing candidate and immune receptors to derive ideas on the potential usefulness and immunogenicity of the construct. Here, the molecular interface between the designed multi-epitope vaccine (MEV) and Toll-like receptor 4 (TLR4), a dire pattern recognition receptor for the innate immune response, was evaluated with the ClusPro server, a powerful and widely used tool for protein–protein docking. The optimized 3D structure of the MEV was utilized as the ligand, whereas the crystal structure of human TLR4 (PDB ID: 3A7Q) was utilized as the receptor. ClusPro produced ten docking models, of which the highest-ranked cluster, comprising 67 members, had the lowest interaction energy of −1428.6 kcal/mol, suggesting a very stable MEV-TLR4 complex. Further interaction analysis by the PDBsum server showed that the MEV construct formed specific and stable interactions with TLR4 Chain A, involving 17 hydrogen bonds, further supporting the structural compatibility and binding affinity of the complex (Figure 6). To assess the thermodynamic viability of the interface between the vaccine and receptor, binding free energy was determined via the PRODIGY web tool. The calculation resulted in a Gibbs free energy (ΔG) of −10.1 kcal/mol, which is equivalent to a dissociation constant (Kd) of 4.1 × 10^−8^ M at 37 °C, reflecting a strong and favorable binding affinity. Cumulatively, all these findings affirm the energetically stable and favorable interaction between the MEV construct and the TLR4 receptor and attest to its ability to initiate efficacious innate immune signaling pathways after administration.

### 2.10. Molecular Dynamics Simulation of the MEV–TLR4 Complex

The effectiveness or stability of the docked vaccine–receptor complex and the dynamic behavior was calculated based on various MD simulation parameters (Figure 7). The RMSD curves of the two chains, i.e., Chain A (receptor) and Chain E (vaccine construct), did not show major drifts in their structure, suggesting that no significant drifts in their structure occurred during the 100 ns trajectory. Chain A had an average amplitude of ~0.6–1.0 nm and Chain E had an average of ~0.4–0.9 nm. Such a steady convergence following the early point of equilibration demonstrates that the complex reached a structural equilibrium in line with stable simulations of the vaccine receptor reported before. (B) Rg values did not change considerably (~2.28–2.36 nm) throughout simulation and this means that the compactness of the complex was maintained. The resultant Rg signature implies little unfolding or unfolding-like conformational changes, indicating the vaccine binding process did not destabilize the overall structural integrity of the receptor, as seen with similar immunoinformatics-based designs of vaccines. RMSF analysis showed that most of the residues that were analyzed in Chain E (C) and Chain A (D) were found to be less flexible. Less significant peaks are associated with both the loop and terminal regions, which are highly flexible and tend to play a critical role in molecular recognition and versatility in protein interactors. The minimal variation in residues in the core binding site corroborates the proposal of constant interchain interactions. (E) The SASA values reduced with time with values of approximately 660 nm 2 at the start and ~550 nm 2 at ~30 ns, settling at that point, reflecting the lower exposure to solvent. This decrease can be attributed to a more compact packing at the surface to which it is binding, implying a stable interaction network between the vaccine and receptor, as is the case in SASA profiles of the high-affinity complex reported. In general, all MD simulation parameters used show the docked vaccine–receptor complex, as indicated, to be structurally stable, dynamically favorable, and compact with the simulation course, which supports it as biologically active.

### 2.11. Normal Mode Analysis of the MEV–TLR4 Complex

To estimate the dynamic behavior, molecular flexibility, and structural stability of the vaccine–receptor complex, normal mode analysis (NMA) was performed with the iMODS server. This study examines the collective motions of biomolecular complexes and gives insights into their deformability and intrinsic flexibility based on internal coordinates. The deformability plot showed minimal distortion over most residues, suggesting structural rigidity within the MEV–TLR4 complex (Figure 8A). Further, B-factor values calculated from NMA showed smaller deviation than experimental B factors, indicating minor atomic fluctuations and supporting the structural rigidity of the complex (Figure 8B). The eigenvalue representing the major mode of motion was 1.655322 × 10^−6^, indicating the rigidity of the complex and the energy needed for distortion. Consistently, the eigenvalues also rose gradually across higher modes, a clear indication of a steady vibrational stiffness pattern (Figure 8C). A plot of variance also showed a reduction in the variances of individual modes, a sign of stable and coordinated inter-residue motion (Figure 8D). In addition, a covariance matrix was plotted to represent correlated motions among pairs of residues. Correlated (red), anti-correlated (blue), and uncorrelated (white) motion patterns were seen, giving an overall picture of the dynamic interaction within the complex (Figure 8E). Concurrently, an elastic network model was developed to show the mechanical stiffness between atomic pairs. Within this model, spring-like bonds were used to represent interatomic interactions, with darker gray dots signifying stronger and more stiff connections (Figure 8F). On average, the NMA outcomes designated that the MEV–TLR4 complex has strong structural stability with low deformability and high internal stiffness, hence confirming the thermodynamic and structural firmness of the vaccine receptor interface under dynamic simulations.

### 2.12. Immune Simulation Analysis

To assess the immunogenic capacity and efficacy of the constructed multi-epitope vaccine (MEV) construct, an in silico immune simulation was performed with the aid of the C-ImmSim server. The outcome of the simulation was a very strong and multi-faceted immune response upon administration of the MEV candidate, which indicated its ability to engage the innate and adaptive limbs of the immune system together. The vaccine induced a vigorous primary immune response, which was marked by a substantial rise in the intensities of immunoglobulins. Of note, the IgM concentration increased dramatically following the first dose, indicative of efficient early immune recognition. IgG1 and IgG2 antibody concentrations were elevated by follow-up doses, while total IgM + IgG responses also increased dramatically, showing efficient class switching development of immunological memory. This demonstrates how the MEV architecture could generate humoral immunity over a long period of time. Furthermore, following numerous antigen exposures, there was also an upsurge in B-cell populations, particularly memory B cells. This proposes the successful development of humoral immunological memory, a component of successful vaccination. Additionally, the simulation revealed a rise in cytotoxic T lymphocytes (CTLs) and helper T lymphocytes (HTLs), primarily in secondary and tertiary immune responses. The sustained decrease in antigen concentrations after the cell-mediated immunity responses demonstrated that the vaccine could neutralize and eradicate the model pathogen. Both adaptive immunity and important elements of the innate immune response were triggered by the vaccine. Following each vaccination, natural killer (NK) cells, macrophages, and proliferation were noted. These innate immune cells are essential for both the early detection of the antigen and the growth of the adaptive immune response. Along with cytokine and interleukin profiles, the immunological simulation also showed increased levels of transforming growth factor-beta (TGF-β) and interferon-gamma (IFN-γ) during the immunization process. Both antiviral defense and immunological modulation depend on these cytokines. Furthermore, other cytokines like interleukin-10 (IL-10), interleukin-23 (IL-23), and interferon-beta (IFN-β) were also present, albeit at lower levels. This suggests a well-balanced cytokine response that reduces excessive inflammation and increases immunogenicity. Throughout the simulation, Simpson’s Diversity Index (D), which gauges the immunological response’s diversity and balance, maintained its ideal value. In order to stop pathogens from escaping through antigenic variation, this test makes sure that the vaccination produces a broad and diverse immune response. These results collectively show that the suggested MEV design can elicit robust, varied, and sustained immune responses, including humoral and cellular immunity. The effectiveness and potential for protection against leishmaniasis of the vaccine are confirmed by the validation of the model’s simulation of immunological dynamics, indicating that it is prepared for additional experimental confirmation (Figure 9).

### 2.13. Codon Adaptation and In Silico Cloning

Codon adaptation and in silico cloning are critical processes to realize effective expression profiling of the vaccine construct within a convenient host, for example, *E. coli*. For this work, codon optimization was conducted with the Java Codon Adaptation Tool (JCat) to convert the codon adaptation pattern of the vaccine sequence to *E. coli* K12 expression. The optimized nucleotide sequence consisted of 397 base pairs, which is equivalent to the multi-epitope vaccine protein designed. CAI of the adjusted sequence was 0.90, and the GC content was 50.71%, both within optimum values (CAI: 0.8–1.0; GC content: 30–70%). These parameters demonstrate a very good potential for high-efficiency transcription and translation within the prokaryotic expression system. The optimized gene sequence of the final multi-epitope vaccine construct was successfully incorporated into the well-consumed pET-28a(+) plasmid vector through in silico cloning. The total size of the obtained recombinant plasmid was confirmed as 4675 base pairs and it ensured accurate insertion of vaccine gene and correctness of the construct for downstream usage in processes like recombinant protein expression and purification from *E. coli* (Figure 10).

## 3. Discussion

When it comes to creating multi-epitope subunit vaccines (MEVs) to combat infectious diseases where conventional methods have failed, immunoinformatics-driven techniques have completely changed the vaccine development process [28]. These methods have confirmed usefulness against a variety of pathogens, such as *Leptospira interrogans*, *Klebsiella pneumoniae*, and *Mycobacterium tuberculosis* [29,30,31]. Here, in the present investigation, we used a systematic immunoinformatics strategy to plan a new and stable MEV targeting the newly discovered Gram-positive bacterium *H. kunzii*, which is increasingly being associated with chronic infections, especially in immunocompromised patients [14].

The proteome of *H. kunzii* underwent subtractive proteomics analysis to detect the most relevant proteins for vaccine development. Among the target proteins identified were peptidoglycan glycosyltransferase FtsW (accession # H3NNL9) and cell division protein FtsZ (H3NNK7), which are crucial to bacterial pathogenicity and survival. FtsW is critical for bacterial cell division and peptidoglycan biosynthesis [32]. Similarly, filamentous temperature-sensitive protein Z (FtsZ), a bacterial counterpart of tubulin, plays a vital role in cell division by polymerizing in a GTP-dependent fashion to assemble the dynamic Z-ring at the future site of septum formation. Given the pivotal role of FtsW and FtsZ in prokaryotic cell division, they have garnered significant attention as a rational and promising target for the development of novel chemotherapeutic agents [33].

Nine CTL, five HTL, and two B-cell epitopes were selected for our investigation based on strict standards, such as significant binding affinity to MHC alleles, immunogenicity, antigenicity, non-toxicity, and non-allergenicity. These epitopes were assessed with a multi-layered in silico pipeline for their broad HLA coverage and immunological significance. T-cell epitope identification is a mainstay of cellular immunity and is crucial in inducing CD8+ cytotoxic and CD4+ helper responses. In this study, CTL epitopes have been chosen based on IC50 values < 2.0 and HTL epitopes with percentile ranks > 2.0, so that they would strongly and broadly interact with MHC molecules. In addition, HTL epitopes were screened for IFN-γ induction, a central cytokine for triggering macrophage activation and improving antigen presentation, like strategies described in prior MEV designs against SARS-CoV-2 [25,34]. Significantly, B-cell epitope mapping, both linear and conformational, guaranteed probable stimulation of humoral immunity. Visualization using PyMOL confirmed their surface accessibility, critical for antigen–antibody interaction [35].

One of the key requirements for epitope-based vaccine design is provision of non-homology to the host proteome in order to minimize the chances of autoimmunity. No considerable homology of the chosen epitopes was found to human proteins, conforming to the primary safety factors outlined in comparable research [36]. Additionally, analysis of population coverage indicated that the combined epitopes cover more than 74% of the world’s population with the greatest coverage in France (88%), Sweden and England (87%), and Germany (86%), indicating worldwide applicability. In order to preserve epitope integrity and improve processing efficiency, the vaccine was methodically created by carefully placing certain epitopes within a framework that includes immunologically suitable linkers (EAAAK, GPGPG, AAY, KK). Due to its potent immunostimulatory properties and proven safety, the cholera toxin B subunit (CTB) was chosen as an adjuvant [37]. This approach is analogous to other research that used CTB and β-defensin to increase systemic and mucosal immune responses [25,38]. With a molecular weight of 43.04 kDa, a pI of 5.56, an instability index of 30.11, and a GRAVY value of −0.068, the vaccine’s physicochemical analysis demonstrated its stability and indicated a hydrophilic, stable, and soluble nature suitable for expression in *E. coli*. Secondary structure analysis revealed an ordered structure with 39.80% coils, 23.68% β-strands, and 36.52% α-helices. A good-quality model was generated using tertiary structure modeling, which was performed with AlphaFold and refined with GalaxyRefine. Validation parameters such as 94% residues in favorable zones (Ramachandran plot), ProSA Z-score = −3.39, and ERRAT quality factor = 86.55 validated its structural accuracy and consistency. Interestingly, more than 60% of vaccine residues were embedded in conformational B-cell epitopes, favoring the potential to elicit strong antibodies, a premise already proven by [39].

A firm interaction with a binding energy of −1428.6 kcal/mol and 17 hydrogen bonds was found when the vaccine design was molecularly docked with human TLR4 (PDB ID: 3A7Q). With a dissociation constant (Kd) of 4.1 × 10^−8^ M and ΔG of −10.1 kcal/mol, the thermodynamic evaluation by PRODIGY exhibited strong and specific interactions with the immune receptor. These findings are consistent with previous studies that had exhibited high-affinity interactions between vaccine constructs and TLR3 or TLR4, resulting in enhanced immunological activation [40,41]. Application of NMA demonstrated positive eigenvalues, high stiffness, and low deformability, reflecting structural rigidity and stability in the MEV-TLR4 complex. These results are in line with previous simulation studies that reported stable interactions between vaccination and TLR [35,41]. Efficient removal of antigens during secondary and tertiary immune responses was the result of substantial IgM and IgG production, B-cell memory development, and significant proliferation of CTLs and HTLs, based on immunological simulation studies. A well-coordinated immune response that strengthens humoral and cellular immunity is indicated by elevated levels of TGF-β, IL−23, and IFN-γ. Simpson’s Diversity Index also confirmed the immunological repertoire’s richness, which is in line with other strong MEV candidates.

Additional research demonstrates the significance of solvent accessibility and thoughtful epitope selection. A parallel MEV study targeting Helicobacter pylori demonstrated that immunodominant, surface-exposed epitopes significantly improved B-cell responses [42]. Our design is based on the idea that the majority of the predicted B-cell epitopes are surface-exposed and enable effective antigen–antibody interactions. Furthermore, prior research on antigen *Leishmania donovani* and *M. tuberculosis* vaccines has shown that adjuvants like β-defensin and CTB are linked to better antigen delivery and presentation [43,44]. Our results support this approach by showing that adding CTB significantly improved antigenicity and structural stability, resulting in a stronger immunological simulation profile. For experimental expression, codon optimization was carried out for *E. coli* K12; a GC content of 50.71% and a CAI of 0.90 were obtained, indicating a high likelihood of effective transcription and translation. Through in silico cloning, the construct was successfully integrated into the pET-28a(+) vector, producing a 5211 bp recombinant plasmid. This result is in line with earlier research that effectively expressed vaccine candidates using comparable vectors [45].

## 4. Materials and Methods

### 4.1. Protein Sequence Retrieval and Selection

The full-scale *H. kunzii* strain ATCC 51366 proteome was retrieved based on the UniProt database, https://www.uniprot.org (accessed on 24 March 2025), which is a free resource with curated and annotated protein sequence data (UniProt ID: UP000004191) [46,47]. The data were obtained in the FASTA format, which represents protein sequences in a plaintext format, providing readability that allows for use in a wide range of bioinformatics tools [47]. The necessary proteins were identified with the help of Geptop 2.0, http://cefg.uestc.edu.cn/geptop/ (accessed on 24 March 2025), thus setting the process of the potential vaccine candidate selection in motion [48]. This practice allowed the proteome to be narrowed down to elements that are essential to the existence of the bacteria and those that can be employed in future studies to aid the development of vaccines [49]. Geptop 2.0 is a sophisticated computational tool that aims to measure the essentiality of proteins using the protein’s sequence information. In this methodology, the functional importance of a particular protein in the biology of the organism is used to identify the difference between essential and non-essential proteins [49]. Essential proteins are very good targets for vaccine formulation as they have a higher probability of being vital for the organism’s pathogenicity and survival [50]. Such proteins’ suitability as vaccine antigens is also augmented by the fact that they often have direct contact with the host immune system [37]. This search facilitates the identification of any potential cross-reactivity by aligning the proteins of *H. kunzii* to the human protein database, https://blast.ncbi.nlm.nih.gov/Blast.cgi (accessed on 25 March 2025) [51]. A widely used method for the comparison of protein sequences that enables one to determine sequence similarities is referred to as the Human Search Tool for Proteins [14,52]. To make sure that only candidates with a low probability of eliciting undesirable immune reactions were considered for further phases of the vaccine development process, proteins with a high-similarity-level sequence homology with human proteins were filtered out before further analysis [35]. PSORTb, https://www.psort.org/psortb/ (accessed on 26 March 2025), was employed to foresee localization of the proteins within the cell that remained after homologous proteins were removed with the use of the BLASTp search [53]. The potential cellular compartments in which the proteins are active are uncovered by this method [54]. Following the exclusion of proteins with any predicted transmembrane regions, transmembrane helices were subsequently evaluated using TMHMM 2.0, https://services.healthtech.dtu.dk/service.php?TMHMM-2.0 (accessed on 26 March 2025) [55]. In order to choose a potential vaccine target, the antigenicity of the proteins that made the shortlist was then assessed employing VaxiJen, http://www.ddg-pharmfac.net/vaxijen/VaxiJen/VaxiJen.html (accessed on 26 March 2025), with a cutoff value set at 0.05 [56,57].

### 4.2. Prediction and Selection of Cytotoxic T-Lymphocyte (CTL) Epitopes

Since they are able to recognize and kill cancerous or infected cells, cytotoxic T lymphocytes (CTLs) contribute significantly to immune protection [58]. In this study, we have used the CTLP red server to make predictions of potential epitopes exhibiting affinity for MHC Class I molecules, http://tools.iedb.org (accessed on 12 April 2025) [59,60]. CTLP red determines epitopes associated with MHC-I molecules in order to trigger the cytotoxic T-cell processes. Epitopes targeted in the development of the vaccine had a very low consensus score, indicating that the epitopes exhibit strong binding affinity to MHC-I alleles [14,60]. By combining the information about sequence-derived content and structure-orientated knowledge, such a method improved the prediction capabilities and made it easier to single out the so-called epitopes, which had a high affinity to MHC-I molecules [61]. After that, the shortlisted candidates were then screened on antigenicity using machine learning-based algorithms on the VaxiJen 2.0 platform, which was utilized to foresee antigenicity [57,62]. The ToxinPred2 https://webs.iiitd.edu.in/raghava/toxinpred2/batch.html (accessed on 12 April 2025) program was consumed to determine the potential harmfulness of these epitopes, whereas AllerTOP 2.0, https://www.ddg-pharmfac.net/AllerTOP (accessed on 12 April 2025), was employed in order to govern the allergenic potential, thus evaluating the safety profile of these epitopes [63,64]. Only those epitopes that had been shown to be non-toxic and non-allergenic entered the last stage of vaccine development to ascertain their compatibility and safety with the immune cells [65].

### 4.3. Prediction of Helper T-Lymphocyte (HTL) Epitopes

The most significant builders of the adaptive immune system are helper T lymphocytes (HTLs) that not only enable humoral and cell-mediated immunity against a wide range of pathogens but also play a highly significant role in protecting against common as well as rare pathogens [66]. The haplotype-transcribed lymphocyte (HTL) epitopes, which can bind to MHC Class II molecules, are of special importance in vaccine development, as they can produce very good immune responses [67,68]. On the basis of their activities in B cells, HTLs play a significant role in producing B cells that are highly specific in attacking and inhibiting antigens of pathogens [15]. Epitopes of HTLs were predicted, including the use of the IEDB MHC Class II binding prediction tool [69] in the current study. Such coordination of a proficient and successful immune response against an invading pathogen requires the concerted efforts of the immune system in a collaborative mechanism with two of its components [70]: stimulation of both macrophages and cytotoxic T lymphocytes (CTLs) by HTLs. How these immune components bind to each other highlights the important contribution of HTL epitopes in designing vaccines since they catalyze an all-inclusive immune response that can recognize, neutralize, and destroy pathogens that are harmful to the body [71].

### 4.4. Prediction and Assessment of B-Cell Epitopes

The epitopes of B cells are vital components of the activation of adaptive immunity, as they trigger antibody-mediated immunity [72]. Designing vaccines using B-cell antigens focuses on B-cell hyper-recognition. The potential B-cell epitopes were foreseen by employing the ABCPred service, https://webs.iiitd.edu.in/raghava/abcpred/ABC_submission.html (accessed on 15 April 2025), which is used to detect linear B-cell epitopes [73]. Since these particular protein regions act as binding zones of antibodies, they become essential in the operations of the adaptive immune system [19]. The ABCPred tool used a cutoff score of 0.5 as a measure to increase the validity of linear B-cell epitope predictions in the current study [74]. Use of this criterion made it probable to choose antibody-recognizable epitopes, which were even more appropriate for integration into the vaccinology system [75].

### 4.5. Construction of the Multi-Epitope Vaccine (MEV) Sequence

The multi-epitope vaccine (MEV) was carefully planned by combining HTL, CTL, and B-cell epitopes as well as suitable peptide connectors and by adding an immune-potentiating adjuvant [76]. To adjust the pace and potency of the immunogenicity-creating effect of the vaccine, a carefully selected adjuvant was incorporated into the formulation [77]. This application was reasoned by the fact that the CTB moiety is characterized by extremely high immunostimulatory activity, https://www.ncbi.nlm.nih.gov/protein (accessed on 16 April 2025) [78]. In order to increase practical integrity and allow functional isolation of the adjuvant and the immunogenic parts, the EAAAK linker was utilized to join the CTB subunit with the N-terminus of the vaccine construct. This type of linker has been proven to make it more stable and guarantee effective spatial separation [14,79]. A number of linkers were prudently chosen in an attempt to preserve the immunogenic potential and functional autonomy of each type of epitope. In particular, AAY peptide linkers were used to display CTL epitopes in such a way that their cytotoxic activity is not impaired, and the GPGPG linkers were employed to display HTL epitopes in order to optimize the helper T-cell stimulatory activity [80]. Putative B-cell epitopes were incorporated via KK links to maintain their antigens’ identity/character and encourage suitable immunological perception [81]. The result was the balance of humoral and cellular immune immunity to achieve the greatest immunological efficiency for the multi-epitope vaccine through this modular design method [82].

### 4.6. Secondary Structure Prediction of the Multi-Epitope Vaccine (MEV)

It was predicted that the secondary structure of the multi-epitope vaccine (MEV) would be composed using the SOPMA and PSIPRED computational programs, https://npsa-prabi.ibcp.fr/cgi-bin/npsa_automat.pl?page=/NPSA/npsa_sopma.html (accessed on 17 April 2025) and http://bioinf.cs.ucl.ac.uk/psipred/ (accessed on 17 April 2025), based on a self-optimized prediction mechanism grounded on sequence alignment [83,84]. SOPMA investigates the natural sequence of the building to extrapolate the structure and arrangement of all key structural components, including coils, long chains, beta sheets, and alpha helices [35]. The fact that the following structural components are examined presents valuable scientific data on the stability and functional structure of the vaccine [85]. This structural prophecy is also key in explaining the possible interactions of the epitopes in the immune system through ensuring that the general conformation of the epitopes stays intact and has the possibility of supporting the most expected immunological effects [86]. The possibility of predicting secondary structure is an important consideration regarding the stability and success of the vaccine construct and forms a crucial piece of information in the continuous functionality of the vaccine’s design development [85].

### 4.7. Tertiary Structure Prediction, Refinement, and Validation of the Multi-Epitope Vaccine (MEV)

Prediction, optimization, and confirmation of the tertiary arrangement of the multi-epitope vaccine (MEV) require the prediction and optimization of the three-dimensional structure because the functionality and stability of a protein are mostly determined by its tertiary form [35]. During the process, the first 3D model was created with the help of the famous AlphaFold 2 server, https://colab.research.google.com/github/sokrypton/ColabFold/blob/main/AlphaFold2.ipynb (accessed on 18 April 2025), which is valued for the accuracy of its protein structure prediction, even of complex multi-domain proteins [87,88]. The GalaxyRefine server, http://galaxy.seoklab.org/cgi-bin/submit.cgi?type=REFINE (accessed on 19 April 2025), was also used to enhance and improve the quality of the structure [89]. The Galaxy Refine server can refine the geometry of the predicted structure through side-chain conformation and general relaxation. The refinement protocol increases the local and global accuracy because it makes the model fit the native protein structure with local and global accuracy [89]. After calibration, the stereochemical quality of the structure was tested with the aid of the Saves v6.1 RAMPAGE tool https://saves.mbi.ucla.edu/ (accessed on 12 April 2025) [90]. This tool also calculates the bond and dihedral angles to determine the chemical and physical viability of the model [91]. Analysis supports the reliability, stability, and satisfaction with the properties of a properly folded protein of the final structure.

### 4.8. Prediction of Discontinuous B-Cell Epitopes

Predicting discontinuous B-cell epitopes is a significant stage in the development of a multi-epitope vaccine (MEV), as the immune structure needs these to identify and bind antibodies [92]. These epitopes were recognized employing the ElliPro algorithm, http://tools.iedb.org/ellipro/ (accessed on 20 April 2025), which predicts discontinuous B-cell epitopes based on the vaccine construct’s 3D conformation [93]. Unlike linear epitopes, discontinuous epitopes are made up of residues that are not near to one another in the primary sequence but are connected within the protein’s tertiary structure [94]. Each predicted epitope is assigned a Protrusion Index (PI) score by ElliPro; higher PI values indicate better surface accessibility and a higher likelihood of antibody interaction [93]. The immune system’s increased awareness of epitope regions with higher PI scores is reflected in the antibody responses [95]. Finding and utilizing these easily accessible epitopes is key to the rational design of vaccines that can efficiently stimulate B cells and increase the likelihood of protective immunity [96].

### 4.9. Molecular Docking of the MEV Construct with TLR4 Receptor

Molecular docking analyses were performed using Toll-like receptor 4 (TLR4) to assess the possible immunogenic interface between human innate immune receptors and multi-epitope vaccination (MEV) [97]. The crystal structure of human TLR4 in complex with MD-2 and its ligand (PDB ID: 3A7Q) served as the docking receptor for simulations in this investigation [98]. The ClusPro server, https://cluspro.bu.edu/login.php (accessed on 21 April 2025), was used to model the interaction between the MEV construct and TLR4. ClusPro uses techniques based on fast Fourier transform (FFT) to predict protein–protein binding conformations that are energetically advantageous [99]. The docking analysis revealed a likely contact that could trigger innate immune signaling pathways. It also revealed crucial details regarding the binding affinity and critical interface residues between the MEV and TLR4 [100]. By stimulating a strong immune response, especially in cases of Gram-positive bacterial infection, this interaction increases the MEV’s potential for efficacy.

### 4.10. Molecular Dynamics Simulation

After the determination of the highest ranked protein–protein docking complex, molecular dynamics simulation was used to optimize and test the stability of the interactions between the vaccine construct designed and the target receptor [101]. The gromacs 2025 suite of software was used for simulation. A solvent cubic box with explicit water molecules enclosed the systems, and sodium and chloride ions with a physiological salt concentration of 150 mM were added to obtain electroneutrality [102]. Energy minimization was performed the same way prior to equilibration, where the steepest descent algorithm was used until maximum force (F max) converged below 10 kJ/mol, ensuring that any form of steric clash would be eliminated, producing a low-energy initial structure [102]. The LINCS algorithm was used to restrain all the covalent bond lengths, and long-range electrostatic interactions were computed via particle mesh Ewald (PME), with a cutoff of 0.9 nm that was applied to both Coulombic and van der Waals energy [103]. The equilibration procedure was carried out in two successive stages, namely the NVT phase (100 ps) to equilibrate the temperature of the system, keeping the number of particles and system volume constant, and then the equilibration of pressure under isothermal–isobaric conditions in the NPT stage (300 ps) [104]. All three spatial dimensions of PBC were enforced to represent an infinite system. Production MD runs were then carried out in equilibrated systems, and data contained in these trajectories were processed with built-in GROMACS utilities [102]. The movement of the molecule and the rough evaluation of the dynamics of interaction were examined using Visual Molecular Dynamics (VMD), and quantitative analyses, such as plotting of the corresponding parameters, were performed in Python 3.13.7 matplotlib [102].

### 4.11. Normal Mode Analysis of Docked Complex

Normal mode analysis was performed to inspect the structural stability and dynamics behavior of the protein–protein complex that formed between the MEV and the TLR4 receptor [105]. In order to obtain even further clarification of the intrinsic motions and flexibility of the MEV-TLR4 complex, normal mode analysis (NMA) was also carried out with the use of the iMODS server, https://imods.iqf.csic.es/ (accessed on 22 April 2025) [106]. Such variables as deformability, B factors, covariance, and eigenvalues were thoroughly researched during the investigation. Through the eigenvalue inspection, the binding of the protein complex was determined to be rigid; the lower the eigenvalues, the more flexible the complex and susceptible it is to a conformation change [107]. To measure the residue-level flexibility of the complex, B factors were calculated, which identified areas of the complex that were subjected to large motion [108]. Covariance analysis revealed very important information on the synergized motions of the complexed subunits of the whole and how critical the motions were for the optimal operation of the vaccine [109]. Deformability assessment was also carried out to establish how certain areas of the complex can change their structure under external stimuli influences [110]. This is a crucial part of the study since it is through it that the manner in which the MEV-TLR4 complex would respond to immune stimulation is likely to be determined, hence giving the vaccination the required qualities of stability and conformational integrity needed to induce a powerful immunological response [35,111]. The results of these simulations offer crucial information regarding the stability, adaptability, and resilience of the vaccine construct—the elements that are vital to the successful deployment of the said structure in vivo.

### 4.12. Immune Response Simulation

An in silico methodology was applied to assess the immunogenicity potential of the multi-epitope vaccine (MEV) construct established through in silico simulation of the immunological reaction conducted with the use of the C-immSim 10.1 server, https://kraken.iac.rm.cnr.it/C-IMMSIM/ (accessed on 23 April 2025) [40]. The computational tool C-ImmSim was specially designed to be able to reproduce immune responses in mammals based on the modeling of interactions within the key immune organs and tissues, especially the lymph nodes and bone marrow [112]. The multi-epitope vaccine (MEV) construct, consisting of protein sequence in the FASTA format, was uploaded to the server to analyze. A set of standard simulation conditions, including single-dose administration (*n* = 1), 10 volumes, 100 simulated step counts, and a fixed random seed number (12,345), was adopted to permit consistency of the simulations [113]. This simulation provides useful predictive information on the interface of the vaccine with the host immune response, including an emulation of humoral immunological effects, cellular immunological effects, and combinations [114]. The process activates memory cells, cytotoxic T lymphocytes, helper T cells, and immunoglobulins. The findings offer initial indications that the MEV is able to elicit a strong immune response and, as such, support the viability of future in vitro and in vivo studies on the same topic [14].

### 4.13. Reverse Translation and Codon Optimization

Reverse translation and codon optimization were performed to promote the successful expression of the multi-epitope vaccine (MEV) in a suitable prokaryotic host with the assistance of the Java Codon Adaptation Tool (JCat) software, http://www.jcat.de (accessed on 24 April 2025) [115]. Such a method assists in renaturing the amino acid sequence of the MEV into a DNA sequence, which is best suited to be expressed in the *E. coli* strain K12 [40]. The Codon Adaptation Index (CAI) was applied in order to measure the suitability of the optimized gene to the preferred usage of codons of the host. An increased CAI value, close to 1.0, reflects better efficiency of the translation and elevated protein expression rates in the host [116]. Moreover, it was also seen that the optimized gene content examined using GC content must be in the possible range (30–70%) to confirm transcriptional stability and facilitate efficient expression [117]. After optimization, in silico cloning of the DNA sequence into the pET28a(+) expression vector was completed by SnapGene program version 3.2.1., https://www.snapgene.com/ (accessed on 25 April 2025) [118]. Because it has essential components that recover the transcription and translation of the inserted gene, the pET28a(+) vector is frequently employed for high-level recombinant protein making in *E. coli* [29]. This phase is critical for making the vaccine construct ready to manufacture on a large scale and subject to additional experimental verification.

## 5. Conclusions

In conclusion, this work offers a thorough immunoinformatics strategy for creating a new multi-epitope subunit vaccine that targets *H. kunzii*. Through the utilization of proteome-scale analyses, antigenicity profiling, molecular docking, and immune simulations, we conceived a construct vaccine that is extremely antigenic, non-allergenic, non-toxic, structurally stable, and competent at eliciting vigorous humoral and cellular immune responses. Interaction with TLR4, combined with high global population coverage and desirable in silico expression parameters, validates the promise of this construct for subsequent experimental validation and preclinical assessment. Although such computational data are encouraging, wet-lab confirmation is essential to conclusively govern the immunoprotective efficacy of the projected vaccine candidate.

## Figures and Tables

**Figure 1 pharmaceuticals-18-01258-f001:**
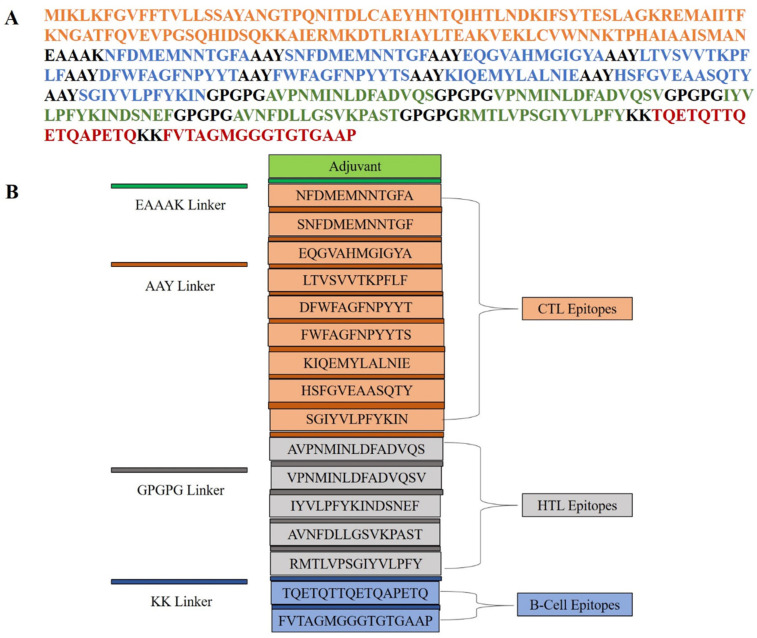
Schematic representation of the multi-epitope vaccine construct. (**A**) Linear amino acid sequence of the designed vaccine, color-coded to indicate the adjuvant (orange), CTL epitopes (blue), HTL epitopes (green), and B-cell epitopes (red). Different linkers (EAAAK, AAY, GPGPG, and KK) are used to join functional domains. (**B**) Modular organization of the vaccine construct, showing the arrangement of adjuvant, CTL, HTL, and B-cell epitopes, each separated by appropriate linkers to ensure proper processing and immunogenicity.

**Figure 2 pharmaceuticals-18-01258-f002:**
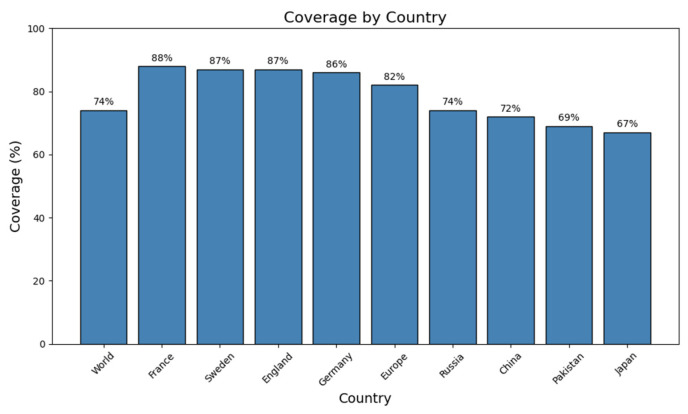
Coverage of the vaccine construct across different countries and regions. Bar graph showing predicted population coverage percentages for the designed vaccine across the world, in selected European countries (France, Sweden, England, and Germany), in Europe overall, and in additional countries (Russia, China, Pakistan, and Japan). France exhibits the highest coverage (88%), followed by Sweden and England (87%), while Japan shows the lowest (67%). These results highlight the broad and variable global coverage potential of the vaccine construct, as commonly reported in immunoinformatics literature.

**Figure 3 pharmaceuticals-18-01258-f003:**
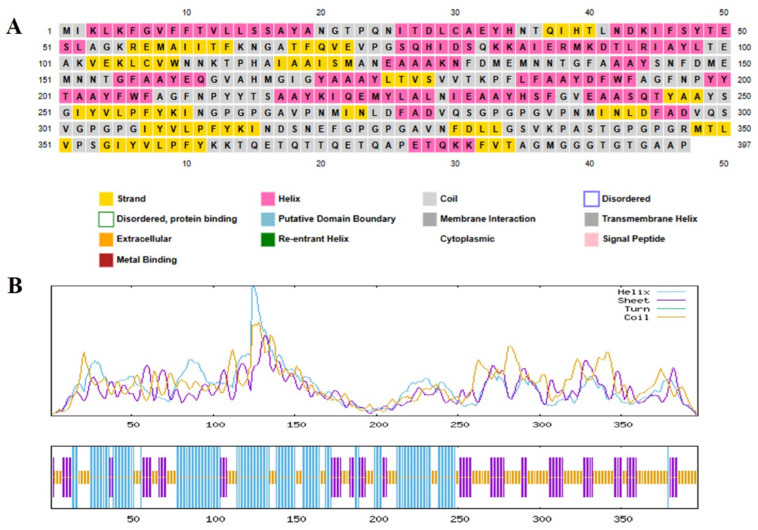
Secondary structure prediction analysis of the target protein sequence. (**A**) PSIPRED prediction results showing the 397-amino-acid sequence with color-coded secondary structure elements: yellow regions represent β-strands, pink areas indicate α-helices, and gray segments denote coil regions. Additional structural features are highlighted according to the legend, including disordered regions, protein-binding sites, putative domain boundaries, membrane interaction regions, cytoplasmic regions, re-entrant helices, transmembrane helices, signal peptides, and metal-binding motifs (**B**) SOPMA secondary structure prediction profile displaying the distribution of structural elements across the protein sequence. The upper graph shows probability scores for helix (purple), sheet (blue), turn (green), and coil (yellow) conformations along the sequence length, with prominent peaks indicating regions of high structural prediction confidence. The lower bar representation provides a simplified visualization of the predicted secondary structure elements at each amino acid position.

**Figure 4 pharmaceuticals-18-01258-f004:**
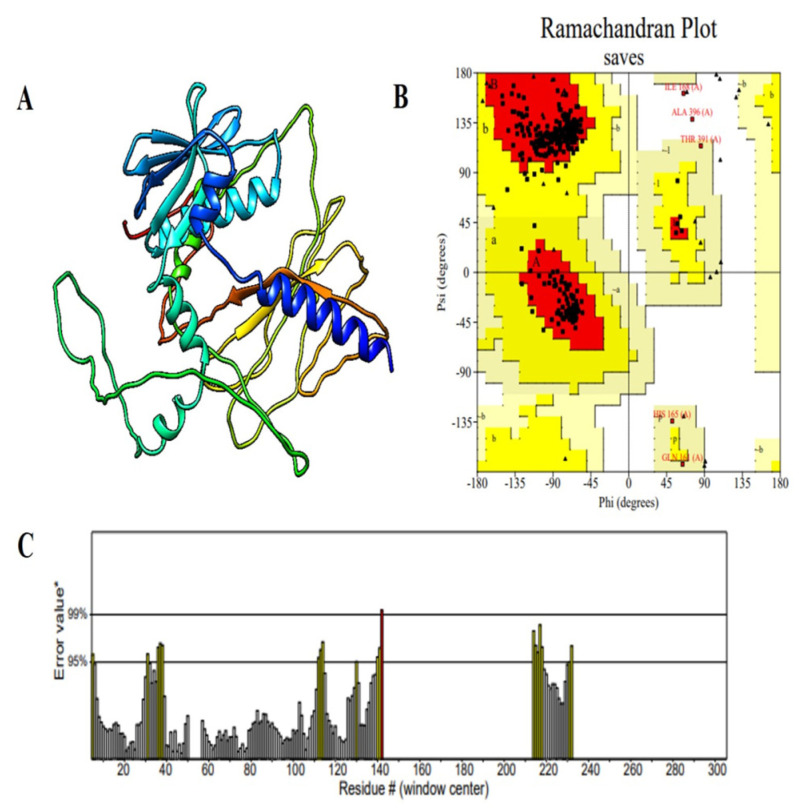
Structural validation of the modeled vaccine construct. (**A**) Cartoon representation of the predicted 3D structure of the vaccine construct, illustrating the arrangement of α-helices, β-strands, and loop regions. (**B**) Ramachandran plot analysis showing the distribution of backbone dihedral angles (φ, ψ) for each residue. Red regions represent the most favored conformations, yellow regions indicate allowed regions, and residues marked in black fall outside the favored regions. (**C**) ERRAT quality factor plot for the vaccine model, displaying the error values for each residue along the sequence. The asterisks (*) represent the 95% and 99% rejection confidence thresholds, while the hash sign (#) denotes the overall ERRAT quality factor score. Most regions exhibit error values below the rejection thresholds, supporting the reliability and stereochemical quality of the predicted structure.

**Figure 5 pharmaceuticals-18-01258-f005:**
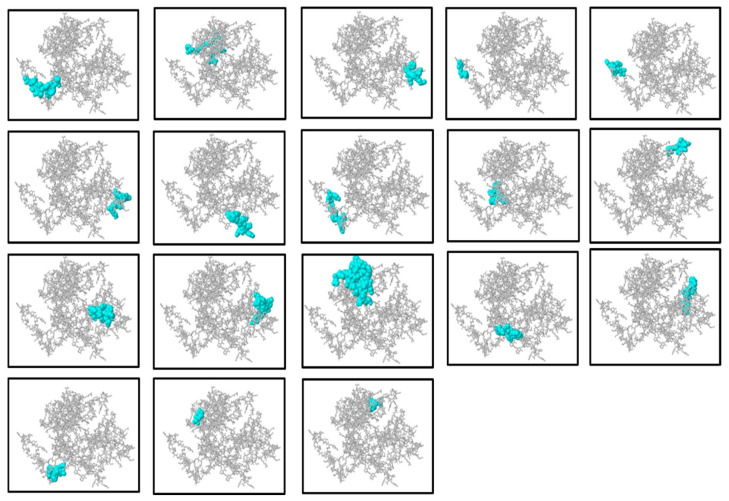
Representation of the discontinuous (conformational) B-cell epitopes inferred by the ElliPro server. The visualization of the three-dimensional construct of the multi-epitope vaccine is illustrated in each panel with the discontinuous epitopes predicted in cyan whereas the rest of the protein is displayed in gray. The cyan regions indicate protrusions and residue clusters predicted to be antigenic and potentially involved in antibody recognition. This spatial mapping helps identify key antigenic sites on the protein surface relevant for immune response. ElliPro uses the protrusions and residue clusters of a protein to predict such epitopes and, based on their spatial adjacency, possible contact with antibodies. This mapping of structure assists in pointing out possible antigenic areas that may be in charge of the recognition of antibodies.

**Figure 6 pharmaceuticals-18-01258-f006:**
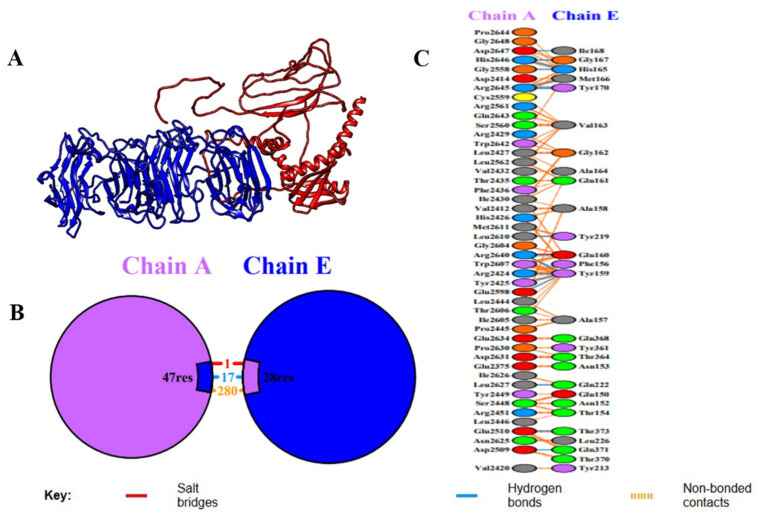
Molecular docking study between the vaccine construct (Chain E) and immune receptor (Chain A). (**A**) Cartoon model of the docked complex, including the space and interaction between the immune receptor (Chain A, blue) and vaccine construct (Chain E, red). The interface of the binding of the two chains is highlighted in the 3D structure. (**B**) Symbolic image of binding interface, which shows the combined number of residues that bind: 47 residues of Chain A (receptor) and 28 residues of Chain E (vaccine construct). The various molecular interactions at the interface are colored differently: salt bridge (red), hydrogen bond (blue), and non-bonded contacts (orange). (**C**) A graphic display of the detailed interactions map, where more specific amino acids residues of Chain A and Chain E participated in the interface. The residues comprising salt bridges, hydrogen bonds, and non-bonded contacts are obviously labeled and color-coded based on the type of interaction, giving us an idea of the molecular dynamics of the vaccine–receptor binding stability.

**Figure 7 pharmaceuticals-18-01258-f007:**
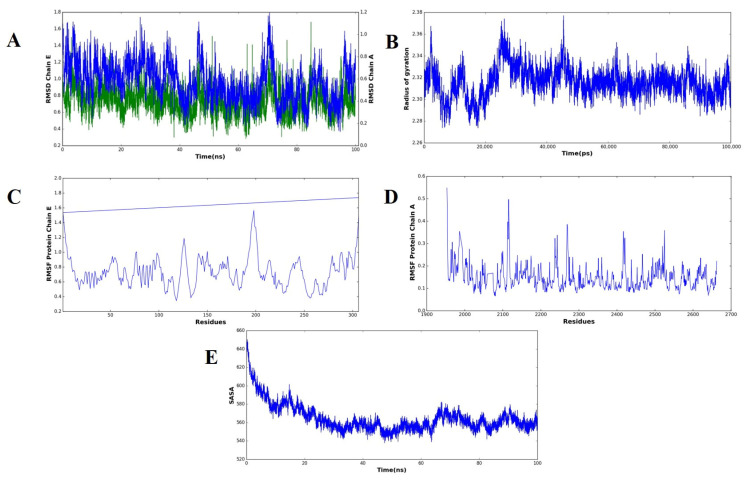
Molecular dynamics simulation of the complex forms of the vaccine and receptor over a 100 ns analysis. (**A**) RMSD plots of structural deviations of Chain A (receptor, blue) and Chain E (vaccine construct, green) that show overall stability after early equilibration. The (**B**) radius of the gyration (Rg) profile demonstrates the uniform tightness of the complex in the simulation. The RMSF plot of Chain E (vaccine construct) can be seen in (**C**) and of Chain A (receptor) in (**D**), showing flexibility, as preferred in loop regions, as well as termini, with core binding residues remaining stable. (**E**) A solvent-accessible surface area (SASA) profile that indicates reduced exposure to solvent with time, indicative of closer packing and complex stability.

**Figure 8 pharmaceuticals-18-01258-f008:**
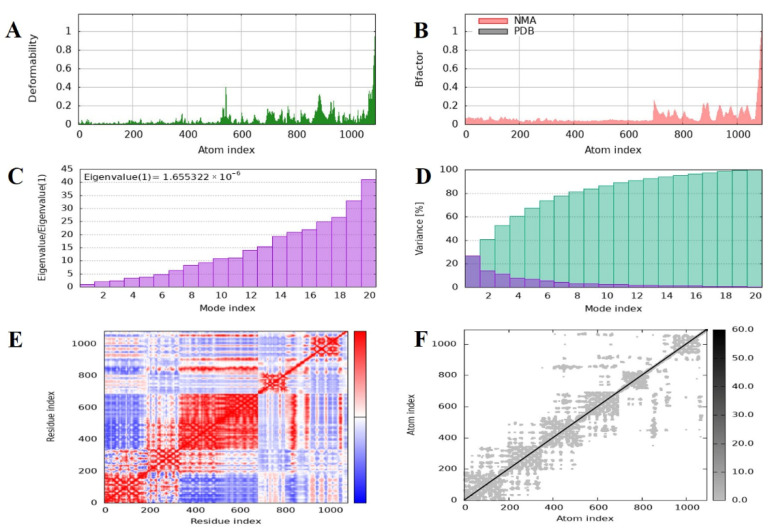
Molecular dynamics simulation and normal mode analysis of the docked vaccine–receptor complex using iMODS. (**A**) Deformability plot showing the flexibility of each residue, with peaks indicating regions of higher mobility within the complex. (**B**) B-factor (mobility) profile comparing the predicted values from normal mode analysis (NMA, red) with those from the PDB structure (black), reflecting atomic fluctuations. (**C**) Eigenvalue graph representing the stiffness of the structure; a lower eigenvalue suggests easier deformation and greater flexibility of the complex. (**D**) Variance plot illustrating the cumulative percentage of motion contributed by each normal mode, with the first few modes accounting for the majority of the motion. (**E**) Covariance matrix map displaying correlated (red), uncorrelated (white), and anti-correlated (blue) motions between residue pairs, indicating dynamic relationships within the complex. (**F**) Elastic network model showing the connections between atoms, where each dot represents a spring between atom pairs, with darker shades indicating stronger interactions.

**Figure 9 pharmaceuticals-18-01258-f009:**
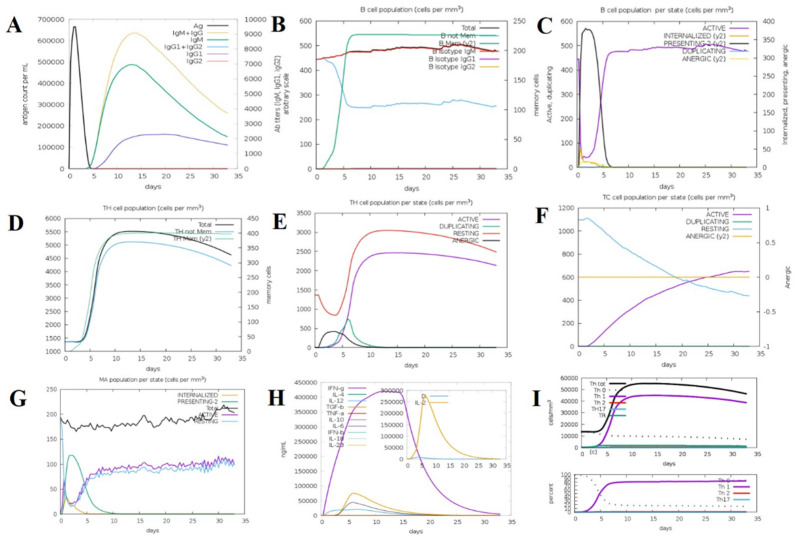
In silico immune response simulation of the designed vaccine using C-ImmSim. (**A**) Antigen and immunoglobulin profile over time, showing rapid antigen clearance and robust production of IgM, IgG, and their subclasses following vaccination. (**B**) B-cell population dynamics, including total B cells, memory B cells, and isotype-specific responses (IgM, IgG1, IgG2), indicating effective induction of memory and isotype switching. (**C**) B-cell state distribution, showing transitions among active, internalized, presenting, duplicating, and anergic states. (**D**) T-helper (TH)-cell population dynamics, with total and memory TH-cell counts over time. (**E**) TH-cell state distribution, highlighting active, duplicating, resting, and anergic TH-cell populations. (**F**) Cytotoxic T-cell (TC) state distribution, showing the proportions of active, duplicating, resting, and anergic TCs. (**G**) Macrophage (MA) state distribution, including internalized, presenting, active, and resting states. (**H**) Cytokine and interleukin profiles, with peak levels of IFN-γ, IL-2, IL-10, TNF-β, and other key cytokines, indicating a strong cellular immune response. (**I**) T-helper-cell subset dynamics, showing the differentiation and abundance of Th0, Th1, Th2, and Th17 subsets over time.

**Figure 10 pharmaceuticals-18-01258-f010:**
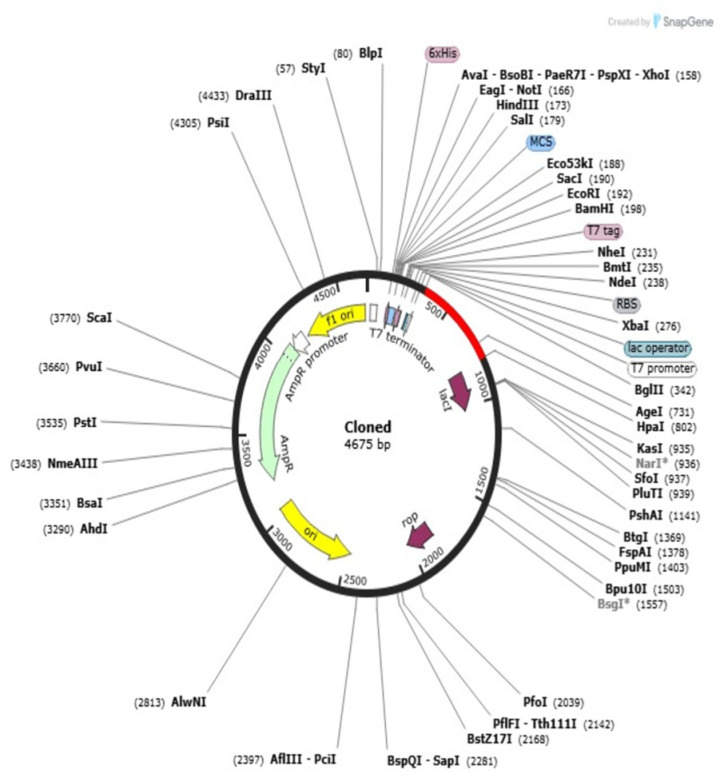
In silico cloning of the optimized gene of the multi-epitope vaccine in expression vector pET28a(+). The circular plasmid map (4675 base pairs) represents the successful insertion and orientation of the codon-optimized vaccine construct into the backbone of pET28a(+) vector. The different arrow colors in the plasmid map represent various vector components: for example, the yellow arrow indicates the promoter region, purple arrows denote antibiotic resistance cassettes, and other colors correspond to cloning sites and inserted vaccine sequences. A detailed legend is provided in the figure to clarify each color. The asterisk (*) in the figure caption indicates the site of successful insertion and orientation of the codon-optimized vaccine construct within the pET28a(+) expression vector. This diagram was created via SnapGene_8.1.1_win.exe software and shows the main characteristics of the selected vectors’ promoter, antibiotic resistance cassette, multiple cloning site, and inserted vaccine sequence. The cloning approach validates the viability of high-level expression of the vaccine protein using an appropriate host system.

**Table 1 pharmaceuticals-18-01258-t001:** Antigenicity, allergenicity, and toxicity predictions for target proteins. Both peptidoglycan glycosyltransferase FtsW (H3NNL9) and cell division protein FtsZ (H3NNK7) show good antigenicity and are predicted to be non-allergenic and non-toxic, supporting their vaccine candidacy.

Accession No.	Protein	Antigenicity	Allergenicity	Toxicity
H3NNL9	Peptidoglycan glycosyltransferase FtsW	0.5169	Non-allergenic	Non-toxic
H3NNK7	Cell division protein FtsZ	0.5439	Non-allergenic	Non-toxic

**Table 2 pharmaceuticals-18-01258-t002:** Predicted CTL epitopes with antigenicity and immunogenicity scores. The table lists cytotoxic T-lymphocyte (CTL) epitopes from FtsZ and FtsW proteins, including their HLA allele restriction, sequence position, antigenicity, and immunogenicity values. These epitopes were selected based on their high antigenicity and favorable immunogenicity, key determinants for effective CTL activation and vaccine design. Note: The asterisk * in HLA allele names (e.g., HLA-B15:02) is part of the standard HLA nomenclature and does not represent a footnote.

Epitope	Protein	Allele	Position	Antigenicity	Immunogenicity
NFDMEMNNTGFA	Cell division protein FtsZ	HLA-B*15:02	3–14	1.1028	−0.12206
SNFDMEMNNTGF	Cell division protein FtsZ	HLA-B*15:02	2–13	0.8996	−0.14401
EQGVAHMGIGYA	Cell division protein FtsZ	HLA-A*26:01	222–233	0.6956	0.13586
LTVSVVTKPFLF	Cell division protein FtsZ	HLA-B*58:01 HLA-B*57:01	129–140	0.6866	−0.16299
DFWFAGFNPYYT	Peptidoglycan glycosyltransferase FtsW	HLA-A*29:02 HLA-B*15:02	767–778	1.3204	0.35278
FWFAGFNPYYTS	Peptidoglycan glycosyltransferase FtsW	HLA-A*29:02 HLA-B*15:02	768–779	1.2422	0.22335
KIQEMYLALNIE	Peptidoglycan glycosyltransferase FtsW	HLA-B*44:03, HLA-B*44:02, HLA-A*32:01, HLA-B*40:02	158–169	1.2288	−0.02229
HSFGVEAASQTY	Peptidoglycan glycosyltransferase FtsW	HLA-A*30:02, HLA-B*57:01, HLA-A*01:01, HLA-B*15:01, HLA-B*35:01	190–201	1.1133	0.05519
SGIYVLPFYKIN	Peptidoglycan glycosyltransferase FtsW	HLA-A*11:01, HLA-A*03:01	439–450	1.0653	0.06436

**Table 3 pharmaceuticals-18-01258-t003:** Predicted HTL epitopes with antigenicity and immunogenicity scores. This table presents helper T-lymphocyte (HTL) epitopes identified from FtsZ and FtsW proteins, along with their HLA-DR allele restrictions, sequence positions, antigenicity, and immunogenicity values. These epitopes were selected based on their high antigenicity and binding affinity, supporting their potential to induce robust CD4+ T-cell responses. Note: The asterisk * in HLA allele names (e.g., HLA-B15:02) is part of the standard HLA nomenclature and does not represent a footnote.

Epitope	Protein	Allele	Position	Antigenicity	Immunogenicity
AVPNMINLDFADVQS	Cell division protein FtsZ	HLA-DRB1*03:06, HLA-DRB1*03:07, HLA-DRB1*03:08	204–218	0.9513	0.09313
VPNMINLDFADVQSV	Cell division protein FtsZ	HLA-DRB1*03:06, HLA-DRB1*03:07, HLA-DRB1*03:08	205–219	0.898	−0.02656
IYVLPFYKINDSNE	Peptidoglycan glycosyltransferase FtsW	HLA-DRB1*04:21, HLA-DRB1*04:26	441–455	1.0957	−0.06595
AVNFDLLGSVKPAST	Peptidoglycan glycosyltransferase FtsW	HLA-DRB1*11:07	288–302	0.8503	−0.25628
RMTLVPSGIYVLPFY	Peptidoglycan glycosyltransferase FtsW	HLA-DRB1*07:03	433–447	0.7638	0.11754

**Table 4 pharmaceuticals-18-01258-t004:** Predicted linear B-cell epitopes with antigenicity and immunogenicity scores. This table lists linear B-cell epitopes from FtsW and FtsZ proteins, including their sequence, position, antigenicity, and immunogenicity values. These epitopes were selected based on high predictive scores, supporting their potential to induce antibody-mediated immune responses.

Epitope	Protein	Score	Position	Antigenicity	Immunogenicity
TQETQTTQETQAPETQ	Peptidoglycan glycosyltransferase FtsW	0.87	1074	1.5134	0.0789
FVTAGMGGGTGTGAAP	Cell division protein FtsZ	0.79	80	1.9827	0.20935

## Data Availability

The data presented in this study are available within the article. No additional datasets were generated or analyzed during the current study.

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

_ANGINOSUS_* Infection via Reverse Vaccinology Approach. Immunology.

