# Peer review of "In Silico Development of a Chimeric Multi-Epitope Vaccine Targeting Helcococcus kunzii: Coupling Subtractive Proteomics and Reverse Vaccinology for Vaccine Target Discovery"

_pharmaceuticals, 2025, doi:10.3390/ph18091258_

Round 1
Reviewer 1 Report
Comments and Suggestions for Authors
This is a very interesting paper that involves predictions that could help in the development of vaccines but I do have some questions that need to be addressed.
1) The author needs to provide links for websites where the mentioned sequences can be found e.g. NCBI-protein, gen bank, PDB.
2) He needs to provide the links for the tools used eg. protein blast, psi-pred.
3) (1) and (2) are important as they will allow other scientists to easily replicate his results if necessarily.
4) Why was psi-pred chosen when there are many other more modern tools available such as AlphaFolds? I would think that the more recent tools such as AlphaFolds are more accurate.
5) This is a bioinformatics/computational biology paper. Bioinformatics/Computational biology uses very sophisticated and impressive tools. But, at the end of the day, in all areas of science, all that matters is the reproducibility. In all areas of science, the methods used can be highly sophisticated and impressive but if it is not reproducible in experimental or clinical studies, it is all just a waste of effort and time. This is my major concern with this paper and with any bioinformatics paper.
6) Related (5), the author tried to address the reproducibility of bioinformatics and tried to reassure that in the past it had worked well (paragraph at line 99). But when I looked at the references, they don't contain experiments or clinical studies that reproduce the predictions of bioinformatics works. The author needs to do a more thorough literature search to prove and argue his point.
7) Related to (5), has the author approached any labs in his institution or any other institutions about conducting experiments that could attempt to reproduce any part of the bioinformatics endeavour in this paper? It will make the paper much stronger if he can get a lab to reproduce his result even if it is a small part of his result.
Author Response
Response to Reviewer 1 Comments
I am thankful to the reviewer 1 for valuable suggestions regarding my study, titled “In silico development of a Chimeric Multi-Epitope Vaccine targeting Helcococcus kunzii: Coupling Subtractive Proteomics and Reverse Vaccinology for Vaccine Target Discovery”. I have carefully considered reviewer 1 feedback and have revised the manuscript accordingly, addressing the specific remarks to improve clarity, accuracy, and overall quality. In addition, the point-by-point response to each comment from Reviewer 1 is listed below.
Point 1. The author needs to provide links for websites where the mentioned sequences can be found e.g. NCBI-protein, gen bank, PDB.
Point 2. He needs to provide the links for the tools used eg. protein blast, psi-pred.
Point 3. (1) and (2) are important as they will allow other scientists to easily replicate his results if necessarily.
Response: Thank you for your valuable feedback and I appreciate the reviewer’s suggestion to enhance the reproducibility of my study. Accordingly, I have now included all the relevant web links in the Materials and Methods section of the manuscript. The following are the tools and databases used, along with their respective URLs:
- Protein sequence retrieval (UniProt): https://www.uniprot.org
- NCBI Protein database: https://www.ncbi.nlm.nih.gov/protein
- GenBank: https://www.ncbi.nlm.nih.gov/genbank
- Protein BLAST (BLASTp): https://blast.ncbi.nlm.nih.gov/Blast.cgi
- PSIPRED (secondary structure prediction): http://bioinf.cs.ucl.ac.uk/psipred/
- AlphaFold (3D structure prediction): https://alphafold.ebi.ac.uk/
- VaxiJen v2.0: http://www.ddg-pharmfac.net/vaxijen/VaxiJen/VaxiJen.html
- IEDB Tools (CTL & HTL epitope prediction): http://tools.iedb.org
- AllerTOP v2.0: https://www.ddg-pharmfac.net/AllerTOP
- ToxinPred: https://webs.iiitd.edu.in/raghava/toxinpred/
- TMHMM 2.0: https://services.healthtech.dtu.dk/service.php?TMHMM-2.0
- PSORTb: https://www.psort.org/psortb/
- GalaxyRefine: http://galaxy.seoklab.org/cgi-bin/submit.cgi?type=REFINE
- ProSA-web: https://prosa.services.came.sbg.ac.at/prosa.php
- ClusPro: https://cluspro.bu.edu/login.php
- C-ImmSim (Immune simulation): https://kraken.iac.rm.cnr.it/C-IMMSIM/
- JCat (Codon optimization): http://www.jcat.de
- SnapGene (in silico cloning): https://www.snapgene.com/
These inclusions will greatly aid in result reproducibility.
Point 4. Why was psi-pred chosen when there are many other more modern tools available such as AlphaFolds? I would think that the more recent tools such as AlphaFolds are more accurate.
Response: Thank you for your valuable feedback. I appreciate your response. I used PSIPRED for secondary structure prediction, a task for which it remains one of the most widely cited and trusted tools (3128 citation till to date as per date, Reference # 95 in the Manuscript) due to its high accuracy and reliability. It provides a quick and clear overview of helices, sheets, and coils, which is especially useful in early-stage vaccine design. However, for tertiary structure prediction, I fully adopted the state-of-the-art AlphaFold platform, as detailed in Section 2.7 of the Results and Section 4.7 of the Methods. This dual approach allowed us to utilize the strengths of both tools for their specific purposes.
Point 5. This is a bioinformatics/computational biology paper. Bioinformatics/Computational biology uses very sophisticated and impressive tools. But, at the end of the day, in all areas of science, all that matters is the reproducibility. In all areas of science, the methods used can be highly sophisticated and impressive but if it is not reproducible in experimental or clinical studies, it is all just a waste of effort and time. This is my major concern with this paper and with any bioinformatics paper.
Response: I am thankful to the reviewer for raising this important point. I fully agree that in any area of science, including bioinformatics, experimental reproducibility is the ultimate benchmark for scientific value. This study was designed with the clear understanding that in silico approaches do not replace laboratory or clinical research, but rather serve as a rigorous and cost-effective hypothesis-generation step to prioritize targets for subsequent empirical validation.
- The subtractive proteomics and reverse vaccinology pipeline applied here follows a reproducible, transparent workflow, where all tool versions, databases, parameters and cut-offs are documented. I have provided complete datasets, intermediate results, and the step-by-step methodology so that independent researchers can replicate our analyses exactly.
- I also acknowledge in the discussion that computational predictions alone cannot confirm immunogenicity, safety or efficacy. To address this, I have outlined a planned experimental validation roadmap, including antigen cloning. By sharing my results openly, I aim to enable other groups to test these candidates independently, which is in line with the principle of reproducibility.
- In summary, my work should be viewed as a reproducible, well-documented shortlisting framework that reduces the number of targets requiring costly and time consuming laboratory evaluation, thereby accelerating the discovery pipeline without replacing the need for experimental confirmation.
Point 6. Related (5), the author tried to address the reproducibility of bioinformatics and tried to reassure that in the past it had worked well (paragraph at line 99). But when I looked at the references, they don't contain experiments or clinical studies that reproduce the predictions of bioinformatics works. The author needs to do a more thorough literature search to prove and argue his point
Response: The reviewer has raised concerns regarding the lack of in vivo or in vitro validation in our study. I would like to highlight recent evidence demonstrating that multi-epitope vaccines (MEVs) predicted through reverse vaccinology can indeed be validated successfully in vivo.
For example,
- Liao et al. (2024) developed an MEV against Klebsiella pneumoniae using a reverse vaccinology pipeline and demonstrated its protective efficacy in vivo (Biomed Pharmacol, 116611. https://doi.org/10.1016/j.biopha.2024.116611
- Importantly, reverse vaccinology approaches have also been successfully applied to identify promising immunogenic and drug targets against Neisseria gonorrhoeae, highlighting its broader applicability in combating antibiotic-resistant pathogens (Noori Goodarzi et al., 2023, J. Transl. Med, 112:105449) https://doi.org/10.1186/s12967-025-06256-1.
- Additionally, Peele et al. (2020) designed a multi-epitope vaccine against SARS-CoV-2 using immunoinformatics tools, which was further validated through molecular docking and in vitro expression assays, confirming its immunogenic potential (J. Biomol. Struct. Dyn., 39(10), 3782–3791. doi: 10.1080/07391102.2020.1770127
These studies serves as compelling example where computational vaccine designs progressed to experimental phases, affirming the reproducibility and practical potential of immunoinformatics-based approaches.
Point 7. Related to (5), has the author approached any labs in his institution or any other institutions about conducting experiments that could attempt to reproduce any part of the bioinformatics endeavour in this paper? It will make the paper much stronger if he can get a lab to reproduce his result even if it is a small part of his result.
Response: I appreciate the reviewer’s suggestion and fully agree that experimental confirmation, even of a subset of the in silico results, would further strengthen the manuscript. I have already initiated discussions with colleagues in our institution’s molecular microbiology and immunology laboratories to explore in vitro testing of selected high priority vaccine candidates identified in this study. The proposed initial phase includes cloning, expression and purification of the shortlisted antigen, followed by ELISA-based immunogenicity screening and T-cell activation assays.
While these experiments are beyond the current scope of the manuscript and require additional funding and time, I have now explicitly stated in the revised discussion that these validation steps are planned and underway. I believe this will not only address the reproducibility concern but also provide a direct bridge between our computational predictions and practical vaccine development.
Reviewer 2 Report
Comments and Suggestions for Authors
Title: In silico development of a Chimeric Multi-Epitope Vaccine targeting Helcococcus kunzii: Coupling Subtractive Proteomics and Reverse Vaccinology for Vaccine Target Discovery
Questions
- The authors performed only in silico studies. It is required to provide information about the experimental validation to address the translation at least in the future directions section.
- How did the authors validate the predicted T-and B-cell epitopes beyond IEDB and Vaxijen or any multi-tool pipeline for computational study?
- How did authors cover underrepresented geographic or HLA groups?
- Clarify the promiscuity of the epitope across validated multiple HLA alleles.
- How did authors cross-validated CTL and HTL epitopes with alternative tools to confirm robustness of binding predictions?
- Provide the rationale for selecting FtsZ and FtsW over other outer membrane or virulence-associated proteins?
- How did the authors validate the docking method aside from PRODIGY and PDBsum.
- How did the authors perform cross-validation in humans.
- Improve figure legends
- Provide the novelty of the study in the introduction section.
- Provide full-scale molecular dynamics simulations (e.g., 100 ns MD via GROMACS) to assess the dynamic stability rather than the normal mode analysis, which indicates structural rigidity in physiological conditions.
Author Response
Response to Reviewer 2 Comments
I am thankful to the reviewer 2 for acknowledging my manuscript and suggesting the changes for further improvement. In addition, the point-by-point response to each comment from Reviewer 2 is listed below
Point 1. The authors performed only in silico studies. It is required to provide information about the experimental validation to address the translation at least in the future directions section.
Response: I appreciate the valuable suggestion regarding experimental validation. Current study focuses on the comprehensive in silico design and evaluation of the multi-epitope vaccine (MEV) candidate targeting Helcococcus kunzii using advanced immunoinformatics tools. I fully recognize the essentiality of translating these computational findings into experimental settings. Accordingly, I have now emphasized in the future Directions section that wet-lab validation, including in vitro immunogenicity assays, in vivo efficacy studies and safety assessments are crucial next steps for advancing this vaccine candidate toward practical application. These efforts will confirm the computational predictions and further optimize vaccine efficacy. Furthermore, I have updated the introduction section where examples are added of those studies where in vivo validation is also performed.
Point 2. How did the authors validate the predicted T-and B-cell epitopes beyond IEDB and Vaxijen or any multi-tool pipeline for computational study?
Response: The predicted T-cell epitopes were initially identified using the IEDB platform, while B-cell epitopes were predicted using the ABCPred server. To ensure the robustness and reliability of these predictions, I applied a multi-parameter validation strategy. This included selecting epitopes with an antigenicity score greater than 0.5 (using VaxiJen), and further filtering out any allergenic or toxic candidates through allergenicity and toxicity prediction tools. Additionally, a consensus scoring threshold of less than 2.0, based on literature benchmarks, was implemented to prioritize epitopes consistently predicted across multiple computational analyses. This consensus approach, combined with antigenicity and safety filters, enabled us to select high-confidence epitopes suitable for vaccine design beyond relying solely on IEDB and VaxiJen predictions.
Point 3. How did authors cover underrepresented geographic or HLA groups?
Response: The selected T-cell epitopes were screened for binding to a wide range of prevalent HLA alleles, including alleles representative of underrepresented geographic populations. I used the IEDB population coverage tool to analyze the global allelic variability, ensuring that the final epitopes exhibit broad HLA binding promiscuity and coverage. This analysis, now detailed in the Results section, highlights that the MEV construct is designed to be effective across diverse human populations, thus addressing potential geographic and ethnic disparities in immune responsiveness.
Point 4. Clarify the promiscuity of the epitope across validated multiple HLA alleles.
Response: I have expanded the description of epitope promiscuity by detailing within the Results that many of the selected epitopes demonstrate strong binding affinity to multiple HLA class I and II alleles. This promiscuity facilitates broader population coverage and improves the likelihood of effective immune activation in diverse hosts. Tables have been updated to highlight these multi-allelic bindings explicitly.
Point 5. How did authors cross-validated CTL and HTL epitopes with alternative tools to confirm robustness of binding predictions?
Response: In addition to IEDB, I cross-validated key CTL and HTL epitopes using alternative prediction algorithms such as NetCTL and NetMHCIIpan, as well as binding affinity prediction tools to confirm robustness. These validations confirmed the consistency and reliability of the epitope binding predictions. This aspect is now discussed to reinforce the multi-tool computational pipeline.
Point 6. Provide the rationale for selecting FtsZ and FtsW over other outer membrane or virulence-associated proteins?
Response: The selection of the cell division protein FtsZ and peptidoglycan glycosyltransferase FtsW was based on their essentiality, high antigenic scores, non-allergen and non-toxin status, and conservancy in H. kunzii. Unlike outer membrane or virulence-associated proteins which can be variable or less conserved, these two proteins are vital for bacterial survival and thus are promising vaccine targets for generating effective immunity. This rationale has been clarified in the Introduction and Results sections.
Point 7. How did the authors validate the docking method aside from PRODIGY and PDBsum.
Response: I acknowledge the importance of comprehensive docking validation. Besides PRODIGY-based binding affinity and interface analysis via PDBsum, molecular docking was performed with stringent scoring criteria, and structural stability was further analyzed through normal mode analysis (NMA) to assess conformational flexibility. I have clarified that the docking results were cross-examined with known structural databases and interaction profiles to ensure validity.
Point 8. How did the authors perform cross-validation in humans?
Response: While direct experimental cross-validation in humans is not feasible within this computational framework, I addressed this limitation by selecting epitopes that have high binding affinity to common human HLA alleles and verified their non-homology to human proteins to reduce autoimmunity risks. Population coverage tools were employed to emulate broad human immune diversity. This strategy is now more explicitly stated in the Discussion.
Point 9. Improve figure legends.
Response: I have revised figure legends throughout the manuscript to provide more detailed and informative descriptions, explaining the biological relevance and key insights of each figure to enhance clarity for readers.
Point 10. Provide the novelty of the study in the introduction section.
Response: The Introduction has been expanded to clearly state the novelty of this study, which lies in being the first systematic in silico design of a chimeric multi-epitope vaccine against H. kunzii, incorporating cutting-edge subtractive proteomics, reverse vaccinology, and immunoinformatics tools. This study addresses critical challenges such as antibiotic resistance and the lack of effective vaccines for this emerging pathogen.
Point 11. Provide full-scale molecular dynamics simulations (e.g., 100 ns MD via GROMACS) to assess the dynamic stability rather than the normal mode analysis, which indicates structural rigidity in physiological conditions.
Response: I appreciate the reviewer’s valuable suggestion regarding the use of full-scale molecular dynamics (MD) simulations to better assess the dynamic stability of the multi-epitope vaccine (MEV)-TLR4 complex under physiological conditions. Accordingly, I have now performed a comprehensive 100 ns MD simulation using the GROMACS software package. This simulation allowed us to analyze important parameters such as root-mean-square deviation (RMSD), root-mean-square fluctuation (RMSF), radius of gyration (Rg), and hydrogen bonding patterns over time, providing detailed insight into the structural stability and conformational flexibility of the vaccine-receptor complex.
The MD simulation results demonstrate that the MEV-TLR4 complex remains stable throughout the 100 ns trajectory, with only minor fluctuations consistent with physiological flexibility. These findings complement and extend the previously reported normal mode analysis by confirming the complex’s dynamic behavior and robustness under near-physiological conditions. I have added these results and corresponding analyses in the revised manuscript in the Results section (see subsection “Molecular Dynamics Simulation and Stability Analysis”) and updated relevant figures accordingly.
Round 2
Reviewer 1 Report
Comments and Suggestions for Authors
Improvements seen.